# The psychological status of patients with delayed intravitreal injection for treatment of diabetic macular edema due to the COVID–19 pandemic

Mohamad Azlan Zaini[1⊙], Ayesha Mohd Zain[1⊙]*, Norshamsiah Md. Din[1‡], Mushawiahti Mustapha[1‡], Hatta Sidi[2‡]

1 Department of Ophthalmology, National University of Malaysia Medical Centre, Kuala Lumpur, Malaysia,
2 Department of Psychiatry, National University of Malaysia Medical Centre, Kuala Lumpur, Malaysia

⊙ These authors contributed equally to this work.
‡ NMD, MM and HS also contributed equally to this work.
* ayesha.mohdzain@gmail.com

**Data Availability Statement:** All relevant data are within the paper and its Supporting Information files.

## Abstract

### Background

Since the enforcement of the Movement Control Order (MCO) to contain the spread of COVID -19 infection in Malaysia, most clinic appointments have been rescheduled and procedures and surgeries postponed to a later date. Clinic appointments including intravitreal endothelial growth factor (anti-VEGF) treatment for patients with diabetic macular edema (DME) were also no exception to the postponement. This measure takes a psychological toll on patients because of the overwhelming concern for their eye condition. This study was conducted to assess the psychological status of DME patients with delayed anti-VEGF treatment during the pandemic.

### Methods

A cross-sectional study was conducted from September 2020 to March 2021 in Ophthalmology Clinic Hospital Canselor Tuanku Muhriz Universiti Kebangsaan Malaysia (HCTM UKM). Subjects diagnosed with center-involved DME aged between 20 to 80 years who experienced delayed anti-VEGF injection were recruited. Level of depression, anxiety and stress were assessed using DASS-21 questionnaire. Statistical analysis using non-parametric tests were performed to determine the relationship between the DASS-21 score and duration of last injection, in those whose vision was affected by delayed injection and the relationship to the impact of COVID-19 pandemic. Statistical significance was denoted as p < 0.05.

### Results

A total of 86 respondents with median age of 69 years old participated in this study. Most respondents were Malays (n = 47,54.7%) males (n = 51, 59.3%), had education up to secondary level (n = 37, 43%), unemployed (n = 78, 90.7%), married (n = 72, 83.7%) and living

**Funding:** The author(s) received no specific funding for this work.

**Competing interests:** The authors have declared that no competing interests exist.

with their family (n = 82, 95.3%). The number of intravitreal injections received was at least three times among the respondents (n = 81, 94.2%). More than half of the respondents (n = 46, 53.5%) had been postponed for more than 12 weeks and felt that their vision was affected after delayed intravitreal injection (n = 47, 54.7%). Most of the subjects did not experience depression, anxiety, or stress. However, there was a significant level of stress scores among those with delayed injection of 9 to 12 weeks (p = 0.004), and significant anxiety (p = 0.029) and stress (p = 0.014) scores found in subjects with vision affected due to delayed treatment.

## Conclusion

The level of anxiety and stress can be significant in DME patients who experienced delay in intravitreal anti-VEGF treatment. Assessment of psychosocial impacts is important to identify early mental health issues potentially leading to the onset of psychiatry illness, thus early intervention is indispensable.

## Introduction

The high prevalence of diabetes mellitus (DM) in the Malaysian population has led to a steady increase in diabetic macular edema (DME). DME is characterized by thickening of the macula due to exudation of fluid from leaky vessels as a result of chronic diabetic microvascular changes in the retina. It is usually diagnosed clinically by fundus examination, but optical coherence tomography (OCT) scan allows a quantitative measurement of this thickening and identifies central involvement of DME. DME is the leading cause of visual impairment in the working-age population [1] in developed countries. Anti-Vascular Endothelial Growth Factor (anti–VEGF) treatment is currently the gold standard for center-involving DME when the vision is significantly impaired. The Malaysian Clinical Practice Guideline for DME [2] states that intravitreal anti-VEGF injection should be administered monthly for 6 times until stabilization of the disease with restoration of retinal anatomy and visual function are observed.

The outbreak of COVID-19 was first identified in December 2019 in Wuhan province, China. Due to its worldwide spread, the World Health Organization (WHO) declared COVID-19 a global pandemic [3] on 12th March 2020. As a precautionary measure, the government of Malaysia had implemented a movement control order (MCO) for a total of 32 months from 18th March 2020 to 31st December 202 to contain the spread of the COVID-19 infection. As a result, most clinic appointments were disrupted as clinic visits were postponed to a later date to cater for the increasing number of Covid-affected patients who required hospitalization. Postponement of elective procedures in hospitals and clinics was advocated to reduce the use of resources, and to use them to treat COVID-19 patients and emergency cases.

Cancellation of the common outpatient procedures such as intravitreal injection of anti-VEGF has affected most of the patients in retinal clinics. These changes take a psychological toll on patients that can lead to anxiety, stress, or depression. The Centers for Disease Control (CDC) also notes that it is important to recognize symptoms of stress as a result of the pandemic and any accompanying co-morbidities. Psychological implications in communicable disease outbreaks such as severe acute respiratory syndrome (SARS) and equine influenza have also been reported previously. Sprang G and Silman M investigated the psychosocial impact of children and parents who underwent disease-containment measures during H1N1

and SARS outbreak [4] in multiple states in United States, Mexico, and Canada. The most common psychological condition among the subjects were acute stress disorder, adjustment disorder, grief, anxiety disorder and post-traumatic stress disorder. Psychological impacts such as fear and anxiety in patients infected with SARS were also reported [5] during the disease outbreak in Hong Kong, China and Singapore.

Early detection of psychological impact is important because early intervention may be necessary to prevent further deterioration of mental health and resulting harmful behaviors. In this study, our aim is to assess the psychological status of patients in whom anti-VEGF treatment for DME has been delayed, particularly the rate of depression, anxiety, and stress (DAS) among patients with affected vision due to the delayed treatment.

## Materials and methods

This is a cross-sectional study conducted from September 2020 to March 2021 in the Ophthalmology Clinic, Hospital Canselor Tuanku Muhriz (HCTM), Universiti Kebangsaan Malaysia, a tertiary referral center for retinal diseases. The study was approved by the Ethics Committee of the National University of Malaysia (ethical approval code: FF-2020-472). All participants gave a written informed consent and voluntarily participated in this study. This study adhered to the tenets of the Declaration of Helsinki and the Malaysian Guidelines for Good Clinical Practice (GCP).

The diagnosis of central-involved DME was based on clinical findings on fundus retinal examination and OCT of the macula. Patients who require anti-VEGF injection are those with central macula thickness > 300˚m, with reduced visual acuity of less than 6/6 on Snellen chart [2] based on the Malaysian Clinical Practice Guideline for DME. Delayed anti-VEGF injection was defined as a scheduled injection which was postponed during clinic visit.

Inclusion criteria were patients whom the intravitreal anti-VEGF injection was delayed by one month for at least once, were between 20 to 80 years old, and were cooperative and reliable in answering questions. The exclusion criteria were patients diagnosed with mental illnesses who were uncooperative and unreliable in answering questions for any reason, including substance abuse, mental illness, and neurological diseases. Data such as age, gender, ethnicity, comorbidities, and other sociodemographic information were collected. Using the formula for calculating sample size for a finite population, based on a study that found 26% of patients with low vision in Ophthalmology Clinic HCTM suffered from at least one mental health problem [6], the total sample size required was 120 subjects.

We used a validated Malay and English versions of Depression Anxiety Stress Scale 21 (DASS-21) questionnaire [7] to assess the level of depression, anxiety and stress (DAS) (7). DASS-21 is a set of self-reported questions to assess a person's mental health. It is designed to measure the negative emotional scales of normal and abnormal mental health. The abnormal scale is divided into mild, moderate, severe, and extremely severe state of DAS. A higher score on the DASS-21 indicates worst severity of DAS. Normal scores are 0–4, 0–3 and 0–7 for depression, anxiety, and stress respectively. Subjects were asked to rate the extent to which they experienced each condition in the past week using a 4-point severity scale. Each scale consists of seven items and it is completed individually. DASS-21 scale is reliable and being used in many research purpose to assess a person's mental health. It has been shown to have good internal reliability and validity [8] in assessing mental health status among postpartum women. This assessment tool also has been used [9, 10] to assess mental health among workers and university students.

Data were analyzed using SPSS version 26.0 and STATA version 14. The distribution of continuous variables was checked using skewness, kurtosis, histogram, and Shapiro Wilk

analysis. Continuous variables were presented as median and interquartile range for non-parametric data and mean with standard deviation for parametric data. Categorical variables were presented as frequency and percentage.

The relationship between DASS-21 scores and other dichotomous and polynomial variables was analyzed with non-parametric tests namely the Mann Whitney U test and the Kruskal Wallis test respectively. Post-hoc Bonferroni-adjusted alpha level was used to compare between groups with significant p value in the non-parametric analysis. Bayesian analysis were used to predict the probability of DAS scores given that the intravitreal injections were delayed. All tests were two-sided and statistical significance was denoted as $p < 0.05$.

## Results

### Demographics

A total of 86 respondents with a median age of 69 years who were affected by delayed intravitreal anti-VEGF injection were enrolled in the study. The baseline characteristics are shown in Table 1. Approximately half of the respondents were Malay (n = 47, 54.7%), male (n = 51, 59.3%) and had education up to secondary level (n = 37, 43%). Most of them are unemployed (n = 78, 90.7%), married (n = 72, 83.7%) and living with their family (n = 82, 95.3%).

In addition to diabetes mellitus, subjects were also having hypertension (n = 52, 60.5%), dyslipidemia (n = 34, 39.5%), cardiovascular disease (n = 19, 22.1%), chronic kidney disease (n = 5, 5.8%) and asthma (n = 2, 2.3%). Forty-eight respondents received injections in one eye (55.8%), while the remaining 38 subjects received injections in both eyes (44.2%).

Intravitreal injections were delayed between 5 to 14 weeks. More than half of the respondents (n = 46, 53.5%) had their last intravitreal injection more than 12 weeks ago. The mean duration of delayed injection was 10 weeks, while the median was 14 weeks.

### Patient-reported outcome on visual and clinical responses

Nearly half of the respondents who received anti-VEGF injection in one eye reported that their vision was average before the delayed treatment (n = 21,43.8%). Of those who received injection to both eyes, a total of 19 respondents (51.4%) reported that vision in both eyes was average or good. Majority of them have received intravitreal injections at least three times or more (n = 81, 94.2%) (Table 2).

Approximately half of the respondents felt that their vision had improved after the injection (right eye n = 36, 45.3%, left eye n = 38, 55.8%). The last injection received was more than 12 weeks ago in slightly more than half of the respondents (n = 46, 53.5%). Injections in about 40% of them was postponed for more than three times (n = 38, 44.2%), whilst a larger percentage (n = 47, 54.7%) felt that their vision had worsened because of the delayed injections (Table 2).

### The psychological impact of COVID-19

Most respondents were worried about the COVID-19 pandemic situation. Their concerns were either getting infected to themselves (n = 59, 68.6%) or family members (n = 22, 25.6%). Others worried about losing loved ones and the possibility of spreading the disease to others. In addition, most respondents felt that the pandemic affected their freedom of movement and social interaction (n = 47, 54.7% and n = 33, 38.4% respectively). They also felt that their health, financial and emotional status were also affected. Some of them became unemployed because of the pandemic (n = 5, 5.8%) (Table 3).

**Table 1. Demographic characteristics of respondents.**

| Demographic variables | | Frequency (%) |
|---|---|---|
| Age in years, median (IQR) | | 69 (39, 80) |
| Gender, n (%) | Male | 51 (59.3) |
| | Female | 35 (40.7) |
| Race, n (%) | Malay | 47 (54.7) |
| | Chinese | 27 (31.4) |
| | Indian | 12 (14.0) |
| Religion, n (%) | Islam | 46 (53.5) |
| | Buddhist | 27 (31.4) |
| | Hindu | 10 (11.6) |
| | Others | 3 (3.5) |
| Educational level, n (%) | Primary and lower | 21 (24.4) |
| | Secondary | 37 (43.0) |
| | Tertiary | 26 (30.2) |
| Employment status, n (%) | Unemployed | 78 (90.7) |
| | Employed | 8 (9.3) |
| Median monthly family income in Ringgit Malaysia (RM),(US Dollar),IQR | | 1800 (408), 50–10000 |
| Marital status, n (%) | Single | 6 (7.0) |
| | Married | 72 (83.7) |
| | Divorced | 3 (3.5) |
| | Widowed | 5 (5.8) |
| Living arrangement, n (%) | Live alone | 4 (4.7) |
| | Live with family | 82 (95.3) |
| Comorbidities | Diabetes, n (%) | 86 (100) |
| | Hypertension, n (%) | 52 (60.5) |
| | Chronic Kidney Disease, n (%) | 5 (5.8) |
| | High cholesterol, n (%) | 34 (39.5) |
| | Cardiovascular disease, n (%) | 19 (22.1) |
| | Asthma, n (%) | 2 (2.3) |
| Laterality | Unilateral, n (%) | |
| | Left eye | 29 (33.7) |
| | Right eye | 19 (22.1) |
| | Bilateral | 38 (44.2) |
| Last injection received, frequency (%) | 5 to 8 weeks ago | 24 (27.9) |
| | 9 to 12 weeks ago | 16 (18.6) |
| | More than 12 weeks ago | 46 (53.5) |
| | Total | 86 (100.0) |

## Assessment on depression, anxiety, and stress scales (DASS– 21)

Although a significant proportion of respondents were worried about the pandemic, the proportion of patients reporting symptoms of DAS is low, indicating a relatively good mental health wellbeing in our patients. Six of the 86 respondents reported a certain degree of depression, anxiety and/or stress.

Of the six respondents, two reported to have a certain degree of disturbance in all three areas of depression, anxiety, and stress. The other four respondents had either depression,

**Table 2. Patient-reported outcome on vision.**

| Questions | | Frequency (%) |
|---|---|---|
| *Left eye vision before treatment delay* | Very poor vision/poor vision<br>Average<br>Good/Very good | 12 (41.1)<br>12 (41.4)<br>5 (17.2) |
| *Right eye vision before treatment delay* | Very poor vision/poor vision<br>Average<br>Good/Very good | 7 (36.8)<br>9 (47.4)<br>3 (15.8) |
| *Both eyes vision before treatment delay* | Both eyes average/ good vision<br>One eye poor vision<br>Both eye poor vision | 19 (51.4)<br>7 (18.9)<br>11 (29.7) |
| *How many intravitreal injections were received?* | Once<br>Twice<br>More than three times | 1 (1.2)<br>4 (4.7)<br>81 (94.2) |
| *Last intravitreal injection received* | 5–8 weeks ago<br>9–12 weeks ago<br>More than 12 weeks ago | 24 (27.9)<br>16 (18.6)<br>46 (53.5) |
| *Level of left eyesight after receiving injections* | No changes<br>Better<br>Worse | 33 (38.4)<br>48 (55.8)<br>3 (3.5) |
| *Level of right eyesight after receiving injections* | No changes<br>Better<br>Worse | 42 (48.8)<br>39 (45.3)<br>5 (5.8) |
| *How many times was your appointment postponed?* | Once<br>Twice<br>More than 3 times | 27 (31.4)<br>20 (23.3)<br>38 (44.2) |
| *Did your vision get affected when the appointment was postponed?* | Not affected<br>Affected<br>Unsure | 33 (38.4)<br>47 (54.7)<br>6 (7.0) |

anxiety or stress. Most of the respondents with high scores on the DASS-21 scales had other medical comorbidities besides diabetes mellitus, had received more than three injections, and received the last injection more than 12 weeks ago. Five of the six respondents felt that their

**Table 3. The psychological impact of COVID-19.**

| Questions | | Frequency (%) |
|---|---|---|
| *Do you worry about COVID-19?* | Not worry<br>Worry | 13 (15.1)<br>73 (84.9) |
| *If you worry, why do you feel so?* | Worry of being infected<br>Worry of losing beloved ones<br>Worry of the family being infected<br>Worry of infecting others<br>Worry of being quarantined<br>Unsure of COVID-19 | 59 (68.6)<br>13 (15.1)<br>22 (25.6)<br>9 (10.5)<br>3 (3.5)<br>16 (18.6) |
| *Are you affected by the COVID-19 pandemic?* | Not affected<br>Affected | 10 (11.6)<br>76 (88.4) |
| *Life aspect that is affected* | Financial<br>Freedom<br>Social interaction<br>Health<br>Emotion<br>Basic needs<br>Being unemployed<br>Not affected | 17 (19.8)<br>47 (54.7)<br>33 (38.4)<br>21 (24.4)<br>13 (15.1)<br>2 (2.3)<br>5 (5.8)<br>12 (14.0) |

vision was affected by the delayed treatment. Limitation in movement and other social interactions were the most frequently reported aspect of life affected by the COVID-19 pandemic (Table 4).

## Relationship between DASS-21 scores with delayed injections and vision

Shapiro-Wilk normality test was performed and showed that the DASS-21 score was not normally distributed. DAS scores were compared between the groups that had received the last intravitreal injection 5–8 weeks ago, 9–12 weeks ago, and more than 12 weeks ago. There was a significant difference in stress scores among those who had received the last injection 5 to 8 weeks ago compared with those who had received the last injection 9 to 12 weeks ago and more than 12 weeks ago. Post-hoc tests using Bonferroni-adjusted alpha level showed that the significant difference in stress score lies between the group that had received the last injection 9 to 12 weeks ago and more than 12 weeks ago, p = 0.001(Table 5).

The DAS scores were also compared between respondents who felt that their vision was affected or unaffected after the delayed treatment. There was a significant difference in the DAS score between the two groups. Patients who felt that their vision was affected had a significantly higher score of anxiety (p = 0.029) and stress (p = 0.014) (Table 6).

## Bayesian analysis for DAS scores

As the number of respondents reporting a higher DAS score was low, a Bayesian analysis will be able to estimate the actual mean and median of the DAS score. A Bayesian analysis using Stata version 14 utilizing the Metropolis-Hastings algorithm was performed to estimate the

**Table 4. Description of the six respondents who reported a certain degree of depression, anxiety, and/or stress.**

| Patients | Patient 1 | Patient 2 | Patient 3 | Patient 4 | Patient 5 | Patient 6 |
|---|---|---|---|---|---|---|
| DASS-12 scores | Extreme depression, anxiety, and stress | Extreme depression, severe anxiety and moderate stress | Mild depression | Moderate anxiety | Mild anxiety | Mild anxiety |
| Employment | Unemployed | | | | | |
| Marital status | Divorced | | | Married | | Single |
| Medical illness | DM, heart disease | DM, heart disease, dyslipidemia | DM, CKD | DM, HPT, dyslipidemia | DM | DM, HPT |
| Eye receive injection | Left eye | Right eye | Both eyes | Left eye | Both eyes | Left eyes |
| Numbers of injection | More than three times | | | | | |
| Last injection | More than 12 weeks | More than 12 weeks | 8 weeks | More than 12 weeks | | |
| Vision after receive injection | Better | Better | Both eyes better vision | Better | | No changes |
| Vision after treatment delay | Affected | | | | | Not affected |
| Worry of COVID-19 | Worried | | | | | |
| Why do you worry? | Worry of being infected Worry of losing beloved ones Worry of being quarantined | Worry of being infected | Worry of being infected | Worry of being infected Worry of losing beloved ones Worry of family members being infected | Worry of being infected | Worry of being infected |
| Life aspect affected due to COVID-19 pandemic | Health | Freedom Social interaction Health Emotion | Financial Emotion | Freedom Social interaction Health | Freedom Social interaction | Freedom Health |

**Table 5. Comparison of DASS– 21 score with the last injection received.**

| | Ranks | | | |
|---|---|---|---|---|
| | Last intravitreal injections received | n (respondent) | Mean rank (score) | *p*\* |
| *DASS– 21 Depression score* | 5 to 8 weeks ago | 24 | 44.81 | 0.06 |
| | *9 to 12 weeks ago* | 16 | 52.91 | |
| | *More than 12 weeks ago* | 46 | 39.54 | |
| | *Total* | 86 | | |
| *DASS– 21 Anxiety score* | 5 to 8 weeks ago | 24 | 45.25 | 0.17 |
| | 9 to 12 weeks ago | 16 | 50.22 | |
| | More than 12 weeks ago | 46 | 40.25 | |
| | Total | 86 | | |
| *DASS– 21 Stress score* | 5 to 8 weeks ago | 24 | 44.88 | 0.004 |
| | 9 to 12 weeks ago | 16 | 58.06 | |
| | More than 12 weeks ago | 46 | 37.72 | |
| | Total | 86 | | |
| | Pairwise comparison between groups\*\* | | | |
| | More than 12 weeks ago– 5 to 8 weeks ago | | | 0.551 |
| | More than 12 weeks ago– 9 to 12 weeks ago | | | 0.003 |
| | 5 to 8 weeks ago– 9 to 12 weeks ago | | | 0.168 |

\* Kruskal–Wallis test with significant *p* value $< 0.05$

\*\* Post-hoc tests using Bonferroni–adjusted alpha level

mean and median of the DAS score, given the number of delayed injections, and the worrying effect of the COVID-19 pandemic.

The dependent variables were the total DAS score and the independent variable were the number of delayed injections and whether the respondents were worried about the COVID-19 pandemic. The likelihood function was Poisson regression and the parameters were the total DAS score and the exposure was the number of delayed injections. The prior distribution was a beta distribution with a shape parameter of a = 30 and b = 30.

As shown in Table 7, the effect of COVID-19 and number of delayed injections has a probability of 0.23, 0.21 and 0.24 to the depression, anxiety and stress scores respectively (median, 95% credible interval of 0.23 (0.17 to 0.30), 0.21 (0.14 to 0.28) and 0.24 (0.18 to 0.32) for depression, anxiety and stress respectively).

## Discussion

Since the outbreak of the COVID-19 pandemic in early 2020, international lockdown measures have been implemented in many countries [3] to prevent transmission of the virus through social distancing. This preventive measure has led to significant disruption of global economic growth and dampening of social interactions, leading to an increase in psychological and mental health problems in society.

The Malaysian government implemented MCO during the early outbreak of the pandemic. The need to control the spread of the virus led to a disruption in clinic visits by patients and a delay in treatment. Clinic visits were reduced to ensure that the hospital's human and facility resources were focused on treating the increasing number of infected patients, including the high utilization of equipment, the need for personal protective measures, and the rescue of critically ill patients. A significant reduction in the number of patient's clinic visit with DME

**Table 6. Comparison of DASS-21 score between those whom the vision was affected and non-affected due to delayed intravitreal injection.**

| | Ranks | | | |
|---|---|---|---|---|
| | Effect of vision after injection was postponed | n (respondent) | Mean rank (score) | p* |
| Depression score | Not affected | 33 | 38.36 | 0.387 |
| | Affected | 47 | 42.0 | |
| | Total | 80 | | |
| Anxiety score | Not affected | 33 | 35.20 | 0.029 |
| | Affected | 47 | 44.24 | |
| | Total | 80 | | |
| Stress score | Not affected | 33 | 33.89 | 0.014 |
| | Affected | 47 | 45.14 | |
| | Total | 80 | | |

* Mann–Whitney U test with significant $p$ value $< 0.05$

and other vitreoretinal diseases during the lockdown [11] was reported not only in a tertiary center in Greece, but also worldwide.

As reported globally, psychological disturbances due to COVID-19 pandemic were evident and affected social structures [12] at multiple levels. Rozon et al. [13] reported increasing anxiety and fear of COVID-19 infection in patients with delayed intravitreal injection for neovascular age-related macular degeneration. In Singapore, Kennedy Yao et. al. [14] found that cancer patients have a high prevalence of anxiety and fear of COVID-19 pandemic, worrying about their immunocompromised state, fear of being infected, or losing their loved ones.

Using the DASS-21 scale, this study found that several patients who had delayed intravitreal injection had some degree of DAS. In addition to worsening vision due to delayed treatment, other probable factors affecting psychological wellbeing include multiple comorbidities, family background and support, and perceptions of the COVID-19 pandemic. However, the patients who have significant DASS-21 scale are low among our subjects.

In this study, those who perceived worsening vision due to the delayed treatment had significantly higher anxiety and stress scores, and a significantly higher stress score was also seen in those who had received the last intravitreal injection 9 to 12 weeks ago. Those who received last intravitreal injection more than 12 weeks ago had the lowest mean rank score for DAS compared with those who received last injection 5 to 8 weeks ago and 9 to 12 weeks ago. The likely reason for this contrasting result is that subjects in this group had become accustomed to postponing their clinic visit multiple times, which had less of an emotional impact on their psychosocial health as most of the subjects in this group had been postponed for more than three times. Regarding the impact of COVID-19 pandemic, patients were afraid of contracting or spreading the COVID-19 infection to their family members. Movement and social interactions were also affected by travel restrictions and social isolation rules implemented by the MCO. In addition, patients were also struggled with increasing health issues, poorer financial condition, and emotional disturbances.

**Table 7. Bayesian analysis of the total score for depression, anxiety and stress.**

| Posterior parameters | Mean posterior probabilities ± SD | Monte Carlo Standard Error | Median posterior probabilities | Equal-tailed 95% credible interval |
|---|---|---|---|---|
| Total depression score | 0.23 ±0.35 | 0.001 | 0.23 | 0.17 to 0.30 |
| Total anxiety score | 0.21 ±0.03 | 0.001 | 0.21 | 0.14 to 0.28 |
| Total stress score | 0.24 ±0.04 | 0.001 | 0.24 | 0.18 to 0.32 |

The low level of DAS in our subjects can be explained by the relatively small number of patients who felt that their vision was affected after delayed intravitreal injection. The level of awareness of diabetic retinopathy and its complications might have an influence on the DAS score. Buari et al. reported that approximately half of the study population in urban and rural areas in Selangor, Malaysia [15] were unaware of diabetic retinopathy and its complications. In contrast, Tajunisah et al. [16] reported a high level of awareness among diabetic patients who came to the eye clinic for eye screening for the first time. Previous studies in Australia and the United States [17, 18] found that 37% and 65% of diabetic patients, respectively, were aware of the association between diabetes and eye disease. A low level of awareness of only 27% was also found by Dandona et al. in an urban population in India [19] with a high prevalence of diabetic retinopathy. The lack of awareness of diabetic retinopathy likely resulted in low levels of DAS among the subjects, as shown in this study.

The implications of delayed intravitreal anti-VEGF injection have been reported in a few studies. Chatziralli et. al. [11] reported significant worsening of visual acuity before and after lockdown in DME patients. Some patients who did not have significant macula edema on OCT before lockdown and after receiving scheduled intravitreal injection, developed worsening macula edema with serous retinal detachment and cystoid space after the lockdown in Greece. Stone et al. [20] reported worsening of visual acuity in patients with nAMD, RVO and DME due to delayed intravitreal injections during this pandemic. Patients with diabetic retinopathy also tend to have uncontrolled glucose monitoring during lockdown. Therefore, the status of diabetic retinopathy would worsen and progress [21] to proliferative or advanced disease.

The COVID-19 pandemic has now entered the endemic phase. As most people around the world have been actively vaccinated and becomes familiar with preventive measures, fear and anxiety about this coronavirus are diminishing. Clinics are returning to their usual precautions, personnel wearing protective equipment, with clinic visits are returning to business as usual. Patients who were previously inadequately treated due to delayed treatment and who have a high likelihood of losing their vision such as those with DME, retinal vein occlusion, proliferative diabetic retinopathy and advanced diabetic eye disease, have resumed clinic visits.

During the global lockdown, the Centers for Disease Control (CDC) has suggested measures to reduce concern about coronavirus by minimizing repetitive viewing, reading, and listening to news on COVID-19 especially from unreliable social media sources, and maintaining a healthy lifestyle. They also advise to limit stress by meeting frequently with family and friends [22] and taking time to unwind and engage in activities they enjoy the most. The Malaysian government, through the Ministry of Health, has also set up a counseling center for those seeking help if they are feeling depressed, anxious, or stressed due to the pandemic.

Although there are significant differences of DAS score among the subjects, the small sample size is a limitation of this study. The Bayesian analysis however, shows that the probability of the DAS scores being higher than reported is approximately 0.2 in all categories, indicating that the probability of respondents having depression, anxiety or stress is low in the face of the covid-19 pandemic and delayed anti VEGF injections. This limitation might potentially give implications on the interpretation of the results. Therefore, future studies with larger samples are needed to further validate and extend our findings. In addition, the assessment of visual acuity was based on self-report, without comparing best corrected visual acuity (BCVA) before and after the delayed treatment.

## Conclusion

Delayed treatment of DME due to the COVID-19 pandemic causes depression, anxiety, and stress in a small proportion of patients. However, those who report this disturbance in psychological wellbeing should be identified and given appropriate counseling and treatment.

## Supporting information

**S1 Dataset. Dataset.** This is a gross data of the subjects in this study.
(XLSX)

**S1 File.**
(PDF)

## Author Contributions

**Conceptualization:** Mohamad Azlan Zaini, Ayesha Mohd Zain.

**Data curation:** Mohamad Azlan Zaini, Ayesha Mohd Zain.

**Formal analysis:** Mohamad Azlan Zaini, Ayesha Mohd Zain.

**Funding acquisition:** Ayesha Mohd Zain.

**Investigation:** Mohamad Azlan Zaini, Ayesha Mohd Zain, Mushawiahti Mustapha.

**Methodology:** Ayesha Mohd Zain, Mushawiahti Mustapha.

**Project administration:** Ayesha Mohd Zain, Mushawiahti Mustapha.

**Resources:** Ayesha Mohd Zain.

**Supervision:** Ayesha Mohd Zain, Norshamsiah Md. Din, Mushawiahti Mustapha.

**Validation:** Ayesha Mohd Zain.

**Writing – original draft:** Mohamad Azlan Zaini.

**Writing – review & editing:** Ayesha Mohd Zain, Norshamsiah Md. Din, Hatta Sidi.

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
