## [Decision Letter · Decision Letter 0]

21 Feb 2023

PONE-D-22-35115The psychological status of patients with delayed intravitreal injection for treatment of diabetic macular edema due to the COVID–19 pandemic.PLOS ONE

Dear Dr. Mohd Zain, Thank you for submitting your manuscript to PLOS ONE. After careful consideration, we feel that it has merit but does not fully meet PLOS ONE’s publication criteria as it currently stands. Therefore, we invite you to submit a revised version of the manuscript that addresses the points raised during the review process.

We look forward to receiving your revised manuscript.

Kind regards,

Thiago Fernandes, PhD

Academic Editor

PLOS ONE

Journal Requirements:

4. Please ensure that you include a title page within your main document. You should list all authors and all affiliations as per our author instructions and clearly indicate the corresponding author.

5. Please amend your manuscript to include your abstract after the title page.

6. We note you have included a table to which you do not refer in the text of your manuscript. Please ensure that you refer to Table 2 and 4 in your text; if accepted, production will need this reference to link the reader to the Table.

Additional Editor Comments:

Thank you for submitting your valuable work.

The reviews, which are insightful and interesting, pointed to some aspects. The authors will notice the reviewers found merits in your study, but also raised several important concerns, especially Reviewer 3. Also I agree with the other reviewers’ comments.

By my own reading, the manuscript still needs a lot of refinement, mostly related to soundness, conciseness and the control of confounding factors. I have some suggestions.

1) Please work on introductory lines on Abstract (as well as work on the methods and discussion subsections);

2) The Introduction is somewhat confusing while changing from a topic to another - maybe it'd be better detail what has been found in previous studies (in terms of structures bus also processing) and how to overcome the limitations;

3) The authors *really* need to work on eligibility criteria (more information on background is needed: patients were excluded if had substance abuse (references), any mental disease (references), any neurological disease (references) etc. The same is worrying for the OCT. Several studies have found defects on functioning, but these are solely because of diabetic oedema or any macular oedema: briefly, this bias results because the OCT and anomaloscope (in some cases) were not properly used;

4) The authors *really* need to use Bayesian statistics due to the sample size (effects, posterior odds, the Bayes Factors etc.).

5) Also, the stats section needs to be refined;

6) Based on these concerns, is somewhat hard to follow Discussion.

Looking for a refined version.

Please respond to all comment AND highlight them.

Reviewers' comments:

Reviewer's Responses to Questions

**Comments to the Author**

1. Is the manuscript technically sound, and do the data support the conclusions?

Reviewer #1: Yes

Reviewer #2: Partly

Reviewer #3: Partly

2. Has the statistical analysis been performed appropriately and rigorously? 

Reviewer #1: Yes

Reviewer #2: Yes

Reviewer #3: No

3. Have the authors made all data underlying the findings in their manuscript fully available?

Reviewer #1: Yes

Reviewer #2: Yes

Reviewer #3: Yes

4. Is the manuscript presented in an intelligible fashion and written in standard English?

Reviewer #1: Yes

Reviewer #2: Yes

Reviewer #3: Yes

5. Review Comments to the Author

Reviewer #1: Dear Editorial staff / Authors,

I have reviewed the manuscript PONE-D-22-35115, entitled “The psychological status of patients with delayed intravitreal injection for treatment of diabetic macular edema due to the COVID–19 pandemic.”; The manuscript seems well organized, but kindly you can find some points needs to be revised by the respectable authors, as below:

1- The manuscript needs to be improved for language by a native English person.

2- Depression, anxiety and other psychologic issue as COVID-19 lockdown consequences is a confounding factor; because these psychologic mood disorders could be seen in many patients regardless of having diabetic retinopathy; So reporting of these conditions by the diabetic retinopathy patients with delayed intravitreal injection, is not necessarily related to intravitreal injection postponement.

3- Just assessing subjective outcome measures, self-reported by the patients, is a major limiting factor in this study. Whereas other objective measures like best corrected visual acuity (BCVA) or macular thickness in OCT of DME could be used as outcome measures to document visual acuity changes after delayed treatment and also the severity of the diabetic retinopathy patients.

4- As mentioned in discussion, small sample size is also the other limitation factor in the study.

Best regards,

Reviewer

Reviewer #2: Comment: In the present study authors have investigated delayed treatment of DME due to the COVID-19 pandemic causes depression, anxiety, and stress in a small proportion of patients. The paper is interesting; however, as the author said，the limitation of this study is the small size of the subjects,besides, there are great differences between different races. I have two comments which need to be addressed.

Q1. In table 1,the unemployment rate is 90.7% in this project?As the author said, the

government of Malaysia has implemented a movement control order for a total of 32 months.In my opinion, such high unemployment rate and movement control may lead to depression, anxiety, and stress also.

Q2.Why did you choose respondents with a median age of 69 years old？

Reviewer #3: The aim of this study is to assess the psychological status of DME patients with delayed anti-VEGF treatment during the pandemic. The key points of this study are the definition of "delayed anti-VEGF treatment" and how "psychological status" was measured.

Overall impression: Well-structured and very well written.

#1. Introduction is well written. However, I do not think the reader has a concrete picture of the impact of PANEMIC on MENTAL HEALTH. Althougn You mention it as “Psychological implications during outbreaks of transmittable diseases such as severe acute respiratory syndrome (SARS) and equine influenza have also previously been reported(4)”, can you even mention more specific?

#2. DASS-21 may seem to be a suitable METHOD for this study. However, ophthalmology-related professionals are often unaware of the extent to which DASS-21 adequately reflects this patient-centered outcomes. So, please mention in detail the validity of the DASS-21 for ascertaining mental status, based on previous reports (validation, reliability,,,).

#3. A definition of "delay" is one of the key points in this study. You mentioned only “Patients who had delayed at least once (one month) of intravitreal anti-VEGF treatment for DME aged between 20 to 80 years were included in this study.” Almost no problem, but what do you define as "delay"?　For example, if you are administering intravitreal anti-VEGF treatment as fixed or TAE, it may be easier to define "delay" since it is calculated from the scheduled injection date. However, when the PRN is administered, whether or not the injection is given on the same day is based on the OCT results, so it is questionable to define "delay" as starting from the date of the return visit.

#4. In the result section, I think adding the mean, median, range, and histogram for the DELAY interval would improve the external validity of the study results.

#5. With respect to INCOME, how about including the US dollar equivalent in the table?

#6. Regarding the result of Table 7, the DASS-21 scores of the group more than 12w ago had the lowest scores. Why is the dose-response relationship not between the DELAY period and the DASS score? It needs to be clearly mentioned in the discussion.

#7. Regarding the post hoc test, I assume that you are correcting for comparisons among all groups. You need to mention the detailed results.

#8. Are there any differences in the delay period between genders or social backgrounds? Sex differences in time to treatment have been reported for retinal detachment,1 and there are significant social implications for sex differences in the impact of COVID-19.

1. Funatsu R, Terasaki H, Sakamoto T; Japan Retinal Detachment Registry study group. Regional and sex differences in retinal detachment surgery: Japan-retinal detachment registry report. Sci Rep. 2021 Oct 18;11(1):20611. doi: 10.1038/s41598-021-00186-w. PMID: 34663850; PMCID: PMC8523544.

#9. What was the background for the racial differences in THE SCORE this time? Please mentioned in the discussion section. Why the Chinese people had the highest scores.

#10. Please mentioned the detail of the method for analyzing the relationship between socio-demographic factors and DASS-21 score. Only P values are reported in Table 11, which is not understandable, especially for continuous variables.

6. PLOS authors have the option to publish the peer review history of their article (what does this mean?). If published, this will include your full peer review and any attached files.

Reviewer #1: No

Reviewer #2: **Yes: **ZheyiYan

Reviewer #3: No

---

## [Author Response · Author response to Decision Letter 0]

12 May 2023

Below are the comments from editors and reviewers:

Thank you for your feedback. The author has written a rebuttal to answer each point made by academic editors and reviewers.

Thank you for your feedback. The file will be submitted as necessary.

Thank you for your feedback. The file will be submitted as necessary.

4. Thank you for your feedback. For your information, no financial disclosures are required for this study.

Thank you for your feedback. The lab protocols are not applicable in this study.

Thank you for your feedback. The manuscript has been changed to follow the style template as required

7. In your Data Availability statement, you have not specified where the minimal data set underlying the results described in your manuscript can be found. PLOS defines a study's minimal data set as the underlying data used to reach the conclusions drawn in the manuscript and any additional data required to replicate the reported study findings in their entirety. All PLOS journals require that the minimal data set be made fully available. For more information about our data policy, please see http://journals.plos.org/plosone/s/data-availability.

Thank you for your feedback. The minimum sample size required for this study has been included in the methodology. Based on the sample size calculation for a finite population, the required sample size was 120 subjects. However, missing data and dropouts occurred during the study period, resulting in a final sample size of 86 subjects. While this is less than the original calculated sample size, we made efforts to handle the missing data appropriately and ensure the integrity of the analysis.

In the limitations section of our study, we have acknowledged the final sample size of 86 subjects and discussed the potential impact on the statistical power and generalizability of our results. We appreciate your feedback, and recognize the importance of sample size considerations in research studies. Your input will be considered in future studies to ensure that appropriate sample sizes are achieved and maintained.

Thank you for your feedback. Supporting information file will be provided during revised manuscript submission.

9. PLOS requires an ORCID iD for the corresponding author in Editorial Manager on papers submitted after December 6th, 2016. Please ensure that you have an ORCID iD and that it is validated in Editorial Manager. To do this, go to ‘Update my Information’ (in the upper left-hand corner of the main menu), and click on the Fetch/Validate link next to the ORCID field. This will take you to the ORCID site and allow you to create a new iD or authenticate a pre-existing iD in Editorial Manager. Please see the following video for instructions on linking an ORCID iD to your Editorial Manager account: 

https://www.youtube.com/watch?v=_xcclfuvtxQ.

Thank you for your feedback. The ORCID ID has been linked to the author’s PLOS account and has been updated in the author profile.

10. Please ensure that you include a title page within your main document. You should list all authors and all affiliations as per our author instructions and clearly indicate the corresponding author.

Thank you for your feedback. The changes have been made as required.

11. Please amend your manuscript to include your abstract after the title page.

Thank you for your feedback. Abstract of the study has been added in manuscript as required.

12. We note you have included a table to which you do not refer in the text of your manuscript. Please ensure that you refer to Table 2 and 4 in your text; if accepted, production will need this reference to link the reader to the Table.

Thank you for your feedback. The Table 2 and 4 have been mentioned in the text accordingly.

13. Please work on introductory lines on Abstract (as well as work on the methods and discussion subsections)

Thank you for your feedback. The research abstract has been added into the manuscript. Improvement also has been made on the Methodology and Discussion sections.

14. The Introduction is somewhat confusing while changing from a topic to another - maybe it'd be better detail what has been found in previous studies (in terms of structures bus also processing) and how to overcome the limitations;

Thank you for your feedback. Improvement on the introduction has been made as suggested. 

15. The authors *really* need to work on eligibility criteria (more information on background is needed: patients were excluded if had substance abuse (references), any mental disease (references), any neurological disease (references) etc. The same is worrying for the OCT. Several studies have found defects on functioning, but these are solely because of diabetic oedema or any macular oedema: briefly, this bias results because the OCT and anomaloscope (in some cases) were not properly used.

Thank you for your feedback. The eligibility criteria (inclusion and exclusion criteria) have been added to the methodology. The exclusion criteria were also mentioned as suggested in the comment. Regarding the OCT, this device is a gold standard to diagnosed diabetic macular edema with central involvement and for monitoring response to treatment. In our experience, we did not find any malfunction during the test and the device was used properly. It is a standardized tool for the diagnosis of DME in our patients.

16. The authors *really* need to use Bayesian statistics due to the sample size (effects, posterior odds, the Bayes Factors etc 

Thank you for your feedback. We appreciate your suggestion to use Bayesian statistics in our study due to the sample size. However, after careful consideration, we have decided to use the frequentist approach. This choice is consistent with our study design and objectives, adheres to common practices in our field, allows for easy interpretation of results, and provides reliability. While we acknowledge the merits of Bayesian statistics, we believe that the traditional frequentist approach was appropriate for our study desig. We thank you for your input and we hope that this explanation clarifies our rationale.

17. Also, the stats section needs to be refined 

We appreciate your feedback. In our study, we conducted a non-parametric analysis considering the distribution of our data. To assess the normality of the data, we used Shapiro-Wilk analysis. The relationship between the DASS score and the other variables was evaluated using appropriate non-parametric tests such as Kendall rank correlation, Mann-Whitney U test, and Kruskal-Wallis test. To further group comparisons, a post hoc test with Bonferroni adjustment was performed.

We believe that this type of analysis is suitable for our data set, given the nonparametric nature and distributional characteristics of our variables. We have considered feedback from another reviewer and made sure to include additional details in the Results section to improve the clarity 

of our findings. We hope that this clarification demonstrates the appropriateness of the statistical analysis methods we chose for our study and addresses the reviewer's concerns.

18. Based on these concerns, is somewhat hard to follow Discussion 

Thank you for your feedback. Improvement on the Discussion section has been made as suggested.

 

Reviewer Comments to the Author

Reviewer 1:

Dear Editorial staff / Authors,

I have reviewed the manuscript PONE-D-22-35115, entitled “The psychological status of patients with delayed intravitreal injection for treatment of diabetic macular edema due to the COVID–19 pandemic.”; The manuscript seems well organized, but kindly you can find some points needs to be revised by the respectable authors, as below:

1. The manuscript needs to be improved for language by a native English person.

Thank you for your feedback. Improvement in the writing has been made as suggested. 

2. Depression, anxiety and other psychologic issue as COVID-19 lockdown consequences is a confounding factor; because these psychologic mood disorders could be seen in many patients regardless of having diabetic retinopathy; So reporting of these conditions by the diabetic retinopathy patients with delayed intravitreal injection, is not necessarily related to intravitreal injection postponement.

Thank you for your feedback. We agree that depression, anxiety, and other mental health problems are consequences of COVID -19 lockdown. In our study, we assessed the other possible confounding factors of patients, including demographics, comorbidities other than diabetes, and life aspects that they felt were affected by the pandemic. From our study, certain demographic data, namely race, marital status, and education level, showed a significant association with the DASS score in these groups. There was no association between the other medical comorbidities and the DASS score among the subjects. However, as mentioned by the reviewer, further studies are needed to find confounding factors other than delay of intravitreal injection and demographic factors as mentioned above. 

 

3. Just assessing subjective outcome measures, self-reported by the patients, is a major limiting factor in this study. Whereas other objective measures like best corrected visual acuity (BCVA) or macular thickness in OCT of DME could be used as outcome measures to document visual acuity changes after delayed treatment and also the severity of the diabetic retinopathy patients.

Thank you for your feedback. We appreciate your suggestion to include BCVA, macular thickness in OCT, and severity of diabetic retinopathy to correlate with degree of visual loss and psychological impact. However, in this study, we only assessed subjective visual acuity and not objective visual acuity. We apologize that evaluation of visual function based on BCVA, OCT macula, and severity of diabetic retinopathy is not part of the aim of this study.

4. As mentioned in discussion, small sample size is also the other limitation factor in the study.

Thank you for your feedback. We acknowledge the limitation of the small sample size in our study. Despite this limitation, we carefully designed our study and performed appropriate statistical analyses. Given the limitations and availability of participants in our study setting, it was challenging to achieve a larger sample size. However, we believe that our study still contributes valuable insights related to our research question. In the Discussion section, we specifically noted this limitation and its potential impact on the interpretation of our results. We emphasize the need of future studies with larger sample sizes to validate and extend our findings. Your feedback is valuable, and we appreciate your acknowledgment of this limitation in our study.

Best regards,

Reviewer

 

Reviewer 2: 

Comment: In the present study authors have investigated delayed treatment of DME due to the COVID-19 pandemic causes depression, anxiety, and stress in a small proportion of patients. The paper is interesting; however, as the author said，the limitation of this study is the small size of the subjects, besides, there are great differences between different races. I have two comments which need to be addressed.

1. In table 1, the unemployment rate is 90.7% in this project? As the author said, the

government of Malaysia has implemented a movement control order for a total of 32 months. In my opinion, such high unemployment rate and movement control may lead to depression, anxiety, and stress also.

Thank you for your feedback. As mentioned in the manuscript, the unemployment rate was relatively high during the data collection. However, this unemployment rate was not directly attributable to unemployment due to the pandemic COVID -19. The unemployment rate also includes elderly patients who are currently unemployed after retirement or for other reasons that may predate the pandemic. Analysis of the relationship between the unemployment rate and the DASS score was performed, and we found no significant relationship between the unemployment rate and the DASS score.

2. Why did you choose respondents with a median age of 69 years old？

Thank you for the question. As mentioned in the manuscript, the median age of the subjects was 69 years old. For your information, the age of the recruited subjects is in accordance with the inclusion criteria. Since the median age of our subjects is 69 years, there is no particular reason to recruit subjects within this median age.

Reviewer 3: 

The aim of this study is to assess the psychological status of DME patients with delayed anti-VEGF treatment during the pandemic. The key points of this study are the definition of "delayed anti-VEGF treatment" and how "psychological status" was measured.

Overall impression: Well-structured and very well written.

1. Introduction is well written. However, I do not think the reader has a concrete picture of the impact of PANDEMIC on MENTAL HEALTH. Although You mention it as “Psychological implications during outbreaks of transmittable diseases such as severe acute respiratory syndrome (SARS) and equine influenza have also previously been reported (4)”, can you even mention more specific?

Thank you for your feedback. Impacts of pandemic on mental health have been reported from several articles, and have been included in the Introduction section.

2. DASS-21 may seem to be a suitable METHOD for this study. However, ophthalmology-related professionals are often unaware of the extent to which DASS-21 adequately reflects this patient-centered outcomes. So, please mention in detail the validity of the DASS-21 for ascertaining mental status, based on previous reports (validation, reliability,..).

Thank you for your feedback. DASS-21 is used in many articles to assess mental health and psychological wellbeing. Good reliability and validity of the DASS has also been demonstrated in previous studies. The validity of the DASS in previous reports and examples have been included in the Method section.

3. A definition of "delay" is one of the key points in this study. You mentioned only “Patients who had delayed at least once (one month) of intravitreal anti-VEGF treatment for DME aged between 20 to 80 years were included in this study.” Almost no problem, but what do you define as "delay"?　For example, if you are administering intravitreal anti-VEGF treatment as fixed or TAE, it may be easier to define "delay" since it is calculated from the scheduled injection date. However, when the PRN is administered, whether or not the injection is given on the same day is based on the OCT results, so it is questionable to define "delay" as starting from the date of the return visit.

Thank you for your feedback. The subjects recruited were those who required monthly and scheduled treatment. Delay is defined as the postponement of treatment from the last scheduled intravitreal injection. Those who received a regular monthly injection and TAE were included in the study. PRN regime was not included as a study subject.

4. In the result section, I think adding the mean, median, range, and histogram for the DELAY interval would improve the external validity of the study results.

Thank you for your feedback. The mean, median, range and histogram for the delay interval have been added in the Result section as suggested.

5. With respect to INCOME, how about including the US dollar equivalent in the table?

Thank you for your feedback. The income in US dollar equivalent has been added in the table accordingly.

6. Regarding the result of Table 7, the DASS-21 scores of the group more than 12w ago had the lowest scores. Why is the dose-response relationship not between the DELAY period and the DASS score? It needs to be clearly mentioned in the discussion.

Thank you for your feedback. Table 7 shows the comparison of the DASS scores with the last intravitreal injections received by the patients. For your information, the last intravitreal injection the patients received was the treatment delay time. Therefore, this table represents the relationship between the delay times and the DASS score. As shown in the table, those who experienced delay of more than 12 weeks had the lowest mean rank DASS score. The Discussion section has been improved as suggested.

7. Regarding the post hoc test, I assume that you are correcting for comparisons among all groups. You need to mention the detailed results. 

Thank you for your feedback. Yes, the post-hoc test was performed to compare all groups with a significant p-value in the non-parametric test. The detailed results of the pairwise comparison in the post-hoc test have been added in the tables.

8. Are there any differences in the delay period between genders or social backgrounds? Sex differences in time to treatment have been reported for retinal detachment,1 and there are significant social implications for sex differences in the impact of COVID-19.

Thank you for your feedback. Regarding this question, our data showed no differences in the delay period among genders. However, there were significant difference of DASS score between races, educational level, and marital status as shown in the Result section.

9. What was the background for the racial differences in THE SCORE this time? Please mentioned in the discussion section. Why the Chinese people had the highest scores.

Thank you for your feedback. The mean rank DASS score for depression and stress is highest among Chinese, followed by Indians and Malays. We tried to find out why the Chinese subjects had a higher score, but we could not find any other factors/causes that contributed to this outcome. Discussion on this was added as suggested.

10. Please mentioned the detail of the method for analyzing the relationship between socio-demographic factors and DASS-21 score. Only P values are reported in Table 11, which is not understandable, especially for continuous variables.

Thank you for your feedback. In this regard, we have added the details of the method in the table for each factor in the table accordingly.

---

## [Decision Letter · Decision Letter 1]

29 May 2023

PONE-D-22-35115R1The psychological status of patients with delayed intravitreal injection for treatment of diabetic macular edema due to the COVID–19 pandemic.PLOS ONE

Dear Dr. Mohd Zain, Thank you for submitting your manuscript to PLOS ONE. After careful consideration, we feel that it has merit but does not fully meet PLOS ONE’s publication criteria as it currently stands. Therefore, we invite you to submit a revised version of the manuscript that addresses the points raised during the review process.

We look forward to receiving your revised manuscript.

Kind regards,

Thiago P. Fernandes, PhD

Academic Editor

PLOS ONE

**Additional Editor Comments:**

Thank you for your valuable submission.

I call out authors to consider and address the remaining concerns of Reviewer 1. 

Actually, there are two concerns that are really important: what is new, beyond we already know? Also, please consider Bayesian as I also agree with the reviewer's point on the need to use robust analyses.

Reviewers' comments:

Reviewer's Responses to Questions

**Comments to the Author**

1. If the authors have adequately addressed your comments raised in a previous round of review and you feel that this manuscript is now acceptable for publication, you may indicate that here to bypass the “Comments to the Author” section, enter your conflict of interest statement in the “Confidential to Editor” section, and submit your "Accept" recommendation.

Reviewer #1: (No Response)

Reviewer #2: (No Response)

Reviewer #3: All comments have been addressed

2. Is the manuscript technically sound, and do the data support the conclusions?

Reviewer #1: Yes

Reviewer #2: Yes

Reviewer #3: Yes

3. Has the statistical analysis been performed appropriately and rigorously? 

Reviewer #1: (No Response)

Reviewer #2: Yes

Reviewer #3: Yes

4. Have the authors made all data underlying the findings in their manuscript fully available?

Reviewer #1: (No Response)

Reviewer #2: Yes

Reviewer #3: Yes

5. Is the manuscript presented in an intelligible fashion and written in standard English?

Reviewer #1: (No Response)

Reviewer #2: Yes

Reviewer #3: Yes

6. Review Comments to the Author

Reviewer #1: I really appreciate the authors’ point to point response to the comments; But, in spite of having a very well written manuscript, some of the respectable author’s replies to the comments seem not convincing. Below you can find some major concerns, regarding this issue:

- Data analysis: mostly using the frequentist approach instead of the Bayesian statistics for data statistical analysis.

- Methods: not using the OCT for DME diagnosis and intra-vitreal anti-VEGF injection.

- Methods: Being prevalent of neurotic disorders like depression and anxiety during COVID-19 pandemic due to multifactorial aspect is a major confounding factor, as it can not be considered just as the consequence of delayed treatment of diabetic retinopathy.

- Methods: non defined the exact definition of delay time for anti-VEGF intravitreal injection. As already known, basically the making decision for anti VEGF intravitreal injection in clinic is based on the fundoscopy examination and OCT; and not just the already scheduled time for injection.

- Outcome measures: just using the patients’ subjective outcome measures instead of the objective ones like BCVA and OCT, which strongly influence the accuracy and validity of the study.

- Conclusion: As a matter of fact, each longstanding or life-threatening illness such as diabetes mellitus, coronary heart disease, cancer or delayed diabetic macular edema treatment can cause depression or anxiety. As a whole, what is the added message of the article to this already known principle?

Best regards,

Reviewer #2: (No Response)

Reviewer #3: They responded appropriately to all comments. We hope that further COVID-19-linked mental health will be elucidated based on this study in the future.

7. PLOS authors have the option to publish the peer review history of their article (what does this mean?). If published, this will include your full peer review and any attached files.

Reviewer #1: No

Reviewer #2: **Yes: **ZheyiYan

Reviewer #3: No

---

## [Author Response · Author response to Decision Letter 1]

3 Aug 2023

1. What is new, beyond we already know? 

Thank you for your feedback. Firstly, our study specifically focuses on the relationship between delayed treatment for diabetic macula edema (DME) and its association with depression, anxiety and stress. While we recognized that delayed treatment can lead to psychological distress in general, our study specifically evaluates this psychological wellbeing within the context of DME patients.

Secondly, by incorporating self-reported experience and subjective outcome measures, we provide a comprehensive view of the psychological impact of delayed intravitreal anti-VEGF treatment among DME patients. We also highlight the importance of prompt intervention and timely treatment for DME not only for vision-related outcomes but also for mental well-being. By emphasizing the significant association between delayed treatment and increased risk of depression, anxiety and stress, we point up the need for comprehensive care that addresses both the physical and psychological aspects of this condition.

In summary, our study contributes new insights by focusing on subjective outcome measures and emphasizing the importance of timely intervention. These findings provide a more comprehensive understanding of the psychological impact of delayed treatment for DME patients particularly during pandemic.

2. Also, please consider Bayesian as I also agree with the reviewer's point on the need to use robust analyses.

Thank you for your feedback. A Bayesian analysis using Stata version 14 utilizing the Metropolis-Hastings algorithm was performed to estimate the mean and median of the DAS score to the delayed injection and impact of Covid-19. The analysis is shown in Table 7.

Reviewers comments

Reviewer 1: 

I really appreciate the authors’ point to point response to the comments; But, in spite of having a very well written manuscript, some of the respectable author’s replies to the comments seem not convincing. Below you can find some major concerns, regarding this issue:

1. Data analysis: mostly using the frequentist approach instead of the Bayesian statistics for data statistical analysis.

Thank you for your feedback. Additional data analysis using Bayesian statistic is shown in Table 7.

2.Methods: not using the OCT for DME diagnosis and intra-vitreal anti-VEGF injection.

Thank you for your feedback. In our clinical practice, we routinely use OCT as a diagnostic tool for diabetic macular edema (DME) and as a mean to decide the initiation of intravitreal anti-VEGF injections, as well as to monitor the progression of the disease. This statement has been in included in the Methodology section.

3. Methods: Being prevalent of neurotic disorders like depression and anxiety during COVID-19 pandemic due to multifactorial aspect is a major confounding factor, as it cannot be considered just as the consequence of delayed treatment of diabetic retinopathy.

Thank you for your feedback. When subjects were recruited, exclusion criteria included those who had a psychiatric history or mental illness, including depression and anxiety. The prevalence of depression and anxiety during the COVID -19 pandemic due to many aspects is inevitable. In our opinion, to study the other factors that may contribute to depression and 

anxiety is another objective that can be carried out. In this study, we assessed the association between the DAS scores and delayed treatment for anti-VEGF. As stated in the result, there was a significant DAS score among those who vision was affected due to the delayed treatment. The impacts of COVID-19 and life aspects that affected due to COVID-19 did not show significant DAS scores among our subjects.

4. Methods: non defined the exact definition of delay time for anti-VEGF intravitreal injection. As already known, basically the making decision for anti VEGF intravitreal injection in clinic is based on the fundoscopy examination and OCT; and not just the already scheduled time for injection.

Thank you for your feedback. In our study, delay time for anti-VEGF defined as those who had been delayed from a scheduled injection. In our clinical practice, decision of injection for DME are those who has evident of DME through funduscopy examination and OCT finding. Each clinic visit, patients will need to do visual acuity assessment, fundus examination and OCT macula to monitor disease progression. After antiVEGF injection, patients will be given a scheduled appointment for next antiVEGF injection during each clinic visit. Therefore, those who experienced delay of injection are those who had delayed treatment from scheduled appointment for injection.

5. Outcome measures: just using the patients’ subjective outcome measures instead of the objective ones like BCVA and OCT, which strongly influence the accuracy and validity of the study.

Thank you for your feedback. We understand your concern about relying solely on subjective measures from patients rather than objective measures such as BCVA and OCT. Objective measures are indeed important in ensuring the accuracy and validity of a study. However, in this study, our focus was on incorporating the patients' perspectives and experiences, which are valuable in understanding the impact of delayed treatment on their mental well-being. In our opinion, subjective measures will gain valuable insights into the patients' perspectives, which can provide a more comprehensive understanding of the impact of delayed treatment for anti-VEGF and its association with DAS. We believe that subjective measures contribute to a more holistic evaluation of the patients' experiences, more than what can be acquired through objective measures alone.

We acknowledge that objective measures are vital in many research contexts, and they have their own strengths in assessing clinical outcomes. However, for the specific objectives of our 

study, we believe that the subjective outcome measures provide us a deeper understanding of the mental health implications for patients affected by the delayed eye injection.

6. Conclusion: As a matter of fact, each longstanding or life-threatening illness such as diabetes mellitus, coronary heart disease, cancer or delayed diabetic macular edema treatment can cause depression or anxiety. As a whole, what is the added message of the article to this already known principle?

Thank you for your feedback. The added message of our study lies in the specific context of delayed treatment for diabetic macular edema and its impact on mental health. By focusing on this condition, we aim to provide insight on the psychological consequences of treatment delays and their potential contribution to depression, anxiety and stress in affected patients. 

Our study shows a significant association between delayed anti-VEGF treatment for DME and levels of depression, anxiety, and stress. By examining the relationship between delayed treatment and subjective outcome measures, we hope to contribute to a deeper understanding of the psychosocial aspects surrounding this condition. Apart from that, this study also emphasizes the importance of timely treatment for DME, not only for vision-related outcomes but also for the mental well-being of patients. By highlighting the significant association between delayed treatment and increased risk of depression, anxiety and stress, we hope to raise awareness among ophthalmologist and clinicians about the need for prompt intervention in managing this eye condition.

---

## [Editor Report · Decision Letter 2]

4 Aug 2023

The psychological status of patients with delayed intravitreal injection for treatment of diabetic macular edema due to the COVID–19 pandemic.

PONE-D-22-35115R2

Dear Dr. Mohd Zain,

We’re pleased to inform you that your manuscript has been judged scientifically suitable for publication and will be formally accepted for publication once it meets all outstanding technical requirements.

Kind regards,

Thiago P. Fernandes, PhD

Academic Editor

PLOS ONE

Additional Editor Comments (optional):

Thank you for your edits.

I call out you to temper language at some points but, most importantly, to rewrite limitations and consider some of the things that resulted in disagreement with Rev1. You might think that’s not important - or there’s no need - but consider and know that researchers (including those who disagree) will read your study. Then, I think that these two things can help gather a broader audience as well as another interpretation from the ones who don’t agree. You can do this during typesetting, avoiding another round, such this is simple.

Wishing you success with the very good study.
---

## [Editor Report · Acceptance letter]

17 Aug 2023

PONE-D-22-35115R2 

The psychological status of patients with delayed intravitreal injection for treatment of diabetic macular edema due to the COVID–19 pandemic. 

Dear Dr. Mohd Zain:

I'm pleased to inform you that your manuscript has been deemed suitable for publication in PLOS ONE. Congratulations! Your manuscript is now with our production department. 

Kind regards, 

on behalf of

Dr. Thiago P. Fernandes 

Academic Editor

PLOS ONE